# *Haemophilus influenzae* Meningitis Direct Diagnosis by Metagenomic Next-Generation Sequencing: A Case Report

**DOI:** 10.3390/pathogens10040461

**Published:** 2021-04-12

**Authors:** Madjid Morsli, Quentin Kerharo, Jeremy Delerce, Pierre-Hugues Roche, Lucas Troude, Michel Drancourt

**Affiliations:** 1IHU Méditerranée Infection, 13005 Marseille, France; mor_madjid@hotmail.com (M.M.); qkerharo@gmail.com (Q.K.); jeremy.delerce@univ-amu.fr (J.D.); 2Aix-Marseille-Université, IRD, MEPHI, IHU Méditerranée Infection, 13005 Marseille, France; 3Department of Neurosurgery, University Hospital of Marseille, 13015 Marseille, France; pierre-hugues.roche@ap-hm.fr (P.-H.R.); lucas.troude@ap-hm.fr (L.T.)

**Keywords:** bacterial meningitis, point-of-care diagnostic, *Haemophilus influenzae*, Oxford Nanopore Technologies, real-time sequencing, metagenomic next-generation sequencing

## Abstract

Current routine real-time PCR methods used for the point-of-care diagnosis of infectious meningitis do not allow for one-shot genotyping of the pathogen, as in the case of deadly *Haemophilus influenzae* meningitis. Real-time PCR diagnosed *H. influenzae* meningitis in a 22-year-old male patient, during his hospitalisation following a more than six-metre fall. Using an Oxford Nanopore Technologies real-time sequencing run in parallel to real-time PCR, we detected the *H. influenzae* genome directly from the cerebrospinal fluid sample in six hours. Furthermore, BLAST analysis of the sequence encoding for a partial DUF417 domain-containing protein diagnosed a non-b serotype, non-typeable *H.*
*influenzae* belonging to lineage *H. influenzae* 22.1-21. The Oxford Nanopore metagenomic next-generation sequencing approach could be considered for the point-of-care diagnosis of infectious meningitis, by direct identification of pathogenic genomes and their genotypes/serotypes.

## 1. Introduction

The microbiological diagnosis of bacterial meningitis presently carried out in point-of-care (POC) laboratories is based on different methods of detection of nucleotide sequences specific to the target pathogen [1,2]. More specifically, this detection is based on techniques using polymerase chain reaction (PCR) amplification by target of pathogen-specific genomic sequences, with the product of the amplification being detected by fluorescence in the so-called real-time PCR (RT-PCR) modality [2,3,4], permitting the detection of the presence or absence of pathogenic genomes, which is not sufficient to carry out pathogen genotyping.

Here, we report one case of *Haemophilus influenzae* meningitis that was diagnosed by metagenomics next-generation sequencing (mNGS), using the technology developed by Oxford Nanopore Technologies (Oxford Nanopore, Oxford Science Park, UK) directly from a cerebrospinal fluid (CSF) sample, identifying the pathogen genome, based on real-time sequencing by comparison to the Oxford Nanopore online database using EP2ME online software.

## 2. Case Report and Methods

A 22-year-old male patient presented to the emergency room of the neurosurgery department at the North Hospital of Marseille following a more than six-metre fall. The patient presented with numerous contusions and a dislocation orbito-naso-ethmoido-frontal type panfacial trauma complicated by subdural haematoma and subarachnoid haemorrhage. At his arrival in the emergency room, the patient presented a Glasgow score of 5, blood pressure of 140/70, cardiac frequency of 110 bpm and 75% saturation in ambient air. Ten days after his admission, the patient presented with sudden onset of febrile (39 °C) meningeal syndrome. Cerebrospinal fluid (CSF) analysis showed a leukocyte count of 198 cell/mm^3^ (37% neutrophils, 63% lymphocytes), erythrocyte count of 4200 cell/mm^3^, protein at 3.31 g/L and glucose at 0.33 mmol/L. Direct microscopic examination after Gram staining did not reveal the presence of any bacteria. A BioFire FilmArray^®^ assay (Biofire bioMérieux, Marcy-l’Etoile, France), performed as previously described [1], yielded *Haemophilus influenzae* according to the meningitis encephalitis panel. The patient was treated with antibiotics for 14 days, 2 days with meropenem (4 g/day) and linezolid (1.2 g/day), 2 days with meropenem only, 2 days with cefotaxime and finally 8 days with amoxicillin at 12 g/day. A drastic decrease in the leukocyte count was observed from the second day of antibiotherapy treatment. The patient suffered cognitive, memory and consciousness disorders and left eye blindness. 

In parallel to the BioFire FilmArray^®^ diagnostic, total DNA was extracted from a 200 μL CSF sample after a 20-minute incubation with proteinase K at 37 °C using an EZ1 DNA Kit (Qiagen, Courtaboeuf, France) and eluted in a 50 µL final volume. For direct microbial genome sequencing, 1 µg of DNA was incorporated into an Oxford Nanopore MinION library preparation according to the manufacturer’s protocol (https://community.nanoporetech.com/protocols/). The library was quantified and normalized in a 47 µL volume using a QubitTM fluorometer using a Qubit dsDNA High Sensitivity Assay Kit (Life Technology, Villebon-sur-Yvette, France). Briefly, DNA repair and end preparation were performed in a 60 µL final volume containing 47 µL of prepared DNA, 1 µL of DNA CS (DNA control), 3.5 µL of NEBNext FFPE DNA Repair buffer, 2 µL of NEBNext FFPE DNA Repair mix (New England BioLabs, Evry-Courcouronnes, France), 3.5 µL of Ultra II End-prep reaction buffer and 3 µL of Ultra II End-prep enzyme mix (New England BioLabs). The repair reaction was performed in a GeneAmp PCR System Thermal Cycler (Applied Biosystems, Foster City, CA, USA) for 5 minutes at 20 °C followed by a 5 min incubation at 65 °C. Repaired DNA was purified using an equal volume of Agencourt Ampure XP beads (Beckman Coulter, Villepinte, France) in the presence of 70% ethanol and then eluted in 61 µL of distilled water. The adapter ligation step was performed in a 100 µL volume containing 60 µL of purified DNA, 25 µL of ligation buffer, 10 µL of T4 DNA ligase (New England BioLabs) and 5 µL of the adapter mix and incubated for 10 minutes at room temperature. A second wash was carried out using Agencourt Ampure XP beads (Beckman Coulter) in a ratio of 0.4. The final library was recovered in a 15 µL final volume, diluted in 75 µL of flow cell loading mix and sequenced for 4 hours on a MinION instrument (Oxford Nanopore). Real-time analysis of sequencing data was performed using EPI2ME software (version 2019.11.11-2920621). A second analysis was performed using Kraken 2 (https://ccb.jhu.edu/software/kraken2/) and visualized by Pavian (https://fbreitwieser.shinyapps.io/pavian/).

## 3. Results and Discussion

After running for 4 h, MinION sequencing generated 202,010 reads including 1277 unclassified reads. Real-time EPI2ME data analysis yielded 192,260 human genome reads, 6598 *Escherichia coli* (control) reads, 29 *Shigella* reads, 23 *Lambdavirus* reads and 11 (0.00005%) reads corresponding to the *H. influenzae* genome. The *Shigella* and *Lambdavirus* reads come from repair and ligation enzymes used in the library preparation. Kraken online analysis and visualization by Pavian online software showed only *H. influenzae* with high stringency (Figure 1). The BLAST analysis of the longest 1298 bp sequence encoding for partial the aerobic respiration control sensor protein *ArcB* gene after specific *H. influenzae* read extraction using Kraken Tools yielded a non-b serotype, non-typeable *H. influenzae* strain P641-4342 with 97% sequence identity (GenBank accession number CP031687.1) belonging to *H. influenzae* lineage 22.1-21 (http://www.iedb.org/sourceOrgId/374927) [5]. This finding has been routinely validated by positive specific *H. influenzae* real-time PCR at 30 Ct using a LightCycler^®^ 480 thermal-cycler (Roche, Wilmington, NC, USA) in a 20 µL final reaction volume containing 5 µL DNA and Takyon No Roxe Probe MasterMix (Eurogentec), targeting a 167 bp fragment length of the *ompP1* gene as previously described [6]. While the patient had been vaccinated for *H. influenzae* serotype B, whole genome sequencing identified a non-typeable *H. influenzae* serotype (NTHi), against which the patient was not immunized. The in silico antibiotic susceptibility pattern derived from the whole genome by ResFinder online software (Version 4.1) predicted *H. influenzae* strain P641-4342 to be susceptible to all antibiotics, as experimentally confirmed by an in vitro antibiogram (Appendix A).

This new approach has been proposed for real-time pathogenic genome detection, essentially based on the time of manipulation of approximately 6 hours in total and its sensitivity to detect microbial genomes, even at low levels. As a novel approach for POC diagnosis, direct identification of the pathogenic genome by metagenomic next-generation sequencing remains a challenge for routine diagnosis. Utilization of Oxford Nanopore technology in mNGS allows real-time sequencing analysis to be performed directly from clinical samples [7]. The *H. influenzae* genome was the first whole bacterial genome to be sequenced [8]. In this case, we diagnosed *H. influenzae* directly from a CSF sample for the first time in our laboratory using Oxford Nanopore Technologies sequencing. Using metagenomics real-time sequencing allowed us to identify an *H. influenzae* non-b serotype lineage 22.1-21 sensitive to all antibiotics by online analysis using ResFinder software in less than 6 hours, which was confirmed by solid culture after three days in routine bacteriology laboratory diagnostics. As a new alternative for POC diagnostics, this strategy will be suitable for implementation in routine diagnostics of CNS infection and genomic surveillance in infectious diseases, despite the variability of genome coverage depending on the pathogenic charge in the CSF.

## 4. Conclusions

Based on the simplicity, rapidity and sensitivity of mNGS real-time sequencing, we are now implementing Oxford Nanopore sequencing technology in the POC laboratory for the rapid diagnosis of bacterial meningitis, providing additional pieces of information over routinely used syndromic real-time PCR kits with which mNGS is competing in terms of delay and cost.

## Figures and Tables

**Figure 1 pathogens-10-00461-f001:**
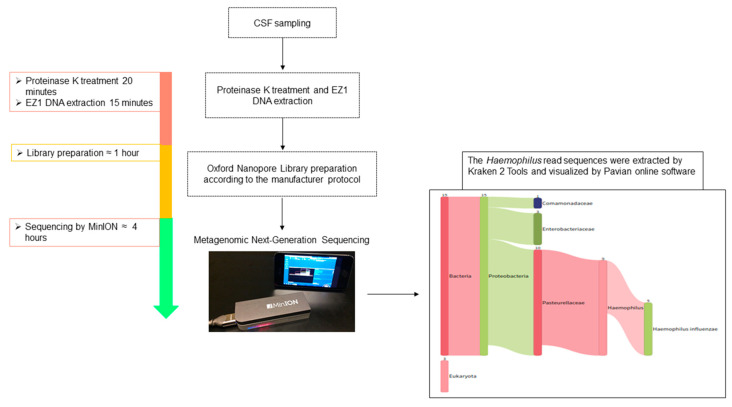
POC real-time sequencing diagnosis of *H. influenzae* meningitis by using Oxford Nanopore Technology. The total process duration was less than 6 h. The cerebrospinal fluid was treated with proteinase K for 20 min at 56 °C, and total DNA was eluted in a 50 µL volume. The Oxford Nanopore library was prepared according to the manufacturer’s protocol and diluted in a 75 µL final volume of flow cell loading mix. The diluted library was sequenced on a MinION instrument, and MinION sequencing data were analysed in real time using EPI2ME software. They were extracted by Kraken 2 Tools and visualized by Pavian online software.

## Data Availability

All the identified genomics sequences have been deposited and are available on the official website of IHU Méditerranée Infection via the following links. https://www.mediterranee-infection.com/acces-ressources/donnees-pour-articles/direct-diagnosis-by-whole-genome-sequencing-of-haemophilus-influenzae-meningitis/. GenBank accession bioproject: PRJNA702049.

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
