# Peer review of "Haemophilus influenzae Meningitis Direct Diagnosis by Metagenomic Next-Generation Sequencing: A Case Report"

_pathogens, 2021, doi:10.3390/pathogens10040461_

Round 1

Reviewer 1 Report

The manuscript by Morsli et al. described the rapid identification of Haemophilus influenzae from CSF in a POC laboratory and using metagenomic NGS technology.

The paper is original and very interesting. The topic is of concern due to the importance of the need to rapidly identify pathogens, their resistance profile in the aim to a best management of antimicrobial agents. This study provides new insight in the routinely use of NGS in the microbiology lab.

I suggest some modifications in the aim to improve the manuscript:

1/ Please add information concerning the case report: give the CRP level; please specify the result of bacterial cultures; give the antibiotic treatment followed (molecule, duration, route of administration, dosage); add the evolution of the patient. Was the patient vaccinated against H. influenzae ?

2/ along the text (Ln 50, 79, 80, 82, 87, 89, 91, 99, 100): Provide bacterial name in italic

3/ Ln 88-89: Please provide the technique use in routine to validate the results with a Ct=30

4/ In conclusion, the authors must develop the interest of the NGS technique compared to Biofire FilmArray.

5/ In appendix 1: Please correct the antibiotic names: amoxicillin/clavulanic acid (not augmentin); gentamicin; nalidixic acid

Moreover could the authors explain that no value for nalidixic acid conducted to the conclusion that the H. influenzae is sensitive to this antibiotic?

Author Response

Response to Reviewer 1 Comments

1/ Please add information concerning the case report: give the CRP level; please specify the result of bacterial cultures; give the antibiotic treatment followed (molecule, duration, route of administration, dosage); add the evolution of the patient. Was the patient vaccinated against H. influenzae?

Authors’ answer: These information’s are now added, "Track Changes" lines 51-55 and lines 98-99.

2/ along the text (Ln 50, 79, 80, 82, 87, 89, 91, 99, 100): Provide bacterial name in italic

Authors’ answer: Corrected accordingly, all bacteria names are in italic.

3/ Ln 88-89: Please provide the technique use in routine to validate the results with a Ct=30

Authors’ answer: Yes indeed, "Track Changes" lines 94-97 along with a new reference [6].

4/ In conclusion, the authors must develop the interest of the NGS technique compared to Biofire FilmArray.

Authors’ answer: Yes indeed, "Track Changes" lines 130-132.

5/ In appendix 1: Please correct the antibiotic names: amoxicillin/clavulanic acid (not augmentin); gentamicin; nalidixic acid

Authors’ answer: Corrected accordingly in the Supplementary data.

6/ Moreover, could the authors explain that no value for nalidixic acid conducted to the conclusion that the H. influenzae is sensitive to this antibiotic?

Authors’ answer: These are standard values used in routine bacteriology recommended by CASFM 30µg/ disc.  

Reviewer 2 Report

Dear Authors, 

This is an interesting and well-written report of the mNGS utilization in the POC testing in meningitis.

I have a question regarding the predicted antibiotic resistance. According to the genomic analysis the detected strain was fully susceptible to the long list of different antibiotics. How likely is that to be true? Isn't it a bit surprising? 

Also, how the cost of the mNGS method compares to the Biofire meningitis panel? 

Line 80 - Should be Shigella, bot Shegilla. 

Kind regards,

(-)

Author Response

Response to Reviewer 2 Comments

1/ I have a question regarding the predicted antibiotic resistance. According to the genomic analysis the detected strain was fully susceptible to the long list of different antibiotics. How likely is that to be true? Isn't it a bit surprising? 

Authors’ answer: In silico susceptibility testing can predict potential antibiotic resistance, but the more important point is to identify susceptibilities and/or resistances to routinely used antibiotics.

2/Also, how the cost of the mNGS method compares to the Biofire meningitis panel? 

Authors’ answer: The average cost of a MinION test is around 220€, but we are currently working to reduce the cost to less than 150€ per test, with the advantage to collect supplementary information about serotype and genotype of the causative bacteria.

3/ Line 80 - Should be Shigella, bot Shegilla.

Authors’ answer: Corrected accordingly, Shegilla replaced by Shigella, "Track Changes" (Lines 85-86).